# Engineering Adenoviral Vectors with Improved GBM Selectivity

**DOI:** 10.3390/v15051086

**Published:** 2023-04-28

**Authors:** Emily A. Bates, Charlotte Lovatt, Alice R. Plein, James A. Davies, Florian A. Siebzehnrubl, Alan L. Parker

**Affiliations:** 1Division of Cancer and Genetics, School of Medicine, Cardiff University, Heath Park, Cardiff CF14 4XN, UK; batese@cardiff.ac.uk (E.A.B.); lovattce@cardiff.ac.uk (C.L.); arplein@gmail.com (A.R.P.); daviesja9@cardiff.ac.uk (J.A.D.); 2European Cancer Stem Cell Research Institute, School of Biosciences, Cardiff University, Maindy Road, Cardiff CF24 4HQ, UK; fas@cardiff.ac.uk; 3Systems Immunity University Research Institute, School of Medicine, Cardiff University, Heath Park, Cardiff CF14 4XN, UK

**Keywords:** brain tumour, oncolytic virus, receptor, therapy

## Abstract

Glioblastoma (GBM) is the most common and aggressive adult brain cancer with an average survival rate of around 15 months in patients receiving standard treatment. Oncolytic adenovirus expressing therapeutic transgenes represent a promising alternative treatment for GBM. Of the many human adenoviral serotypes described to date, adenovirus 5 (HAdV-C5) has been the most utilised clinically and experimentally. However, the use of Ad5 as an anti-cancer agent may be hampered by naturally high seroprevalence rates to HAdV-C5 coupled with the infection of healthy cells via native receptors. To explore whether alternative natural adenoviral tropisms are better suited to GBM therapeutics, we pseudotyped an HAdV-C5-based platform using the fibre knob protein from alternative serotypes. We demonstrate that the adenoviral entry receptor coxsackie, adenovirus receptor (CAR) and CD46 are highly expressed by both GBM and healthy brain tissue, whereas Desmoglein 2 (DSG2) is expressed at a low level in GBM. We demonstrate that adenoviral pseudotypes, engaging CAR, CD46 and DSG2, effectively transduce GBM cells. However, the presence of these receptors on non-transformed cells presents the possibility of off-target effects and therapeutic transgene expression in healthy cells. To enhance the specificity of transgene expression to GBM, we assessed the potential for tumour-specific promoters hTERT and survivin to drive reporter gene expression selectively in GBM cell lines. We demonstrate tight GBM-specific transgene expression using these constructs, indicating that the combination of pseudotyping and tumour-specific promoter approaches may enable the development of efficacious therapies better suited to GBM.

## 1. Introduction

Glioblastoma (GBM), a tumour thought to arise from neuroglial stem or progenitor cells, is the most aggressive primary adult brain cancer. It has an average incidence of 3 cases per 100,000 people per year worldwide [1], making it the most common type of malignant adult brain neoplasm. The current standard treatment procedure for GBM involves the maximum surgical resection of the tumour (where possible), followed by radiation therapy and concomitant chemotherapy using the oral alkylating agent temozolomide (TMZ). Nevertheless, due to its heterogeneity, invasiveness, and rapid growth, the prognosis for GBM patients is very poor, where the mean overall survival is only approximately 15 months [2].

Oncolytic viruses, which are viruses engineered to selectively infect and lyse tumour cells, constitute a promising alternative therapeutic agent for GBM (reviewed in [3]). Their ability to self-amplify within tumour cells not only expands the therapy at the point of need (i.e., within the tumour microenvironment, TME), but also induces immunogenic cell death (ICD) through the lytic nature of cell death. The power of oncolytic viruses can be further augmented via the engineering of potent therapeutic transgenes into the viral genome that can further enhance immune activation and immune-cell-mediated tumour cell killing. Collectively, these traits have the potential to instigate an anti-glioma immune response against both the primary tumour and metastatic growth within the brain.

Whilst the array of viruses available for oncolytic applications are wide, adenoviruses (HAdV) have proven to be the most popular as oncolytic viruses against several types of cancer, as gauged by the volume of clinical trials conducted in the area to date [4]. They have a clinically proven safety profile, are relatively amenable to genetic manipulation and are able to accommodate relatively large transgene inserts [5]. However, to date, the promising results demonstrated in vitro and in murine models of cancer have largely failed to translate into significant efficacy in clinical trials. Whilst these trials have demonstrated feasibility and safety, improvements in cancer patient outcomes have generally been modest [6,7]. Limited efficacy may be due in part to the over reliance of many clinical studies on adenovirus 5 (HAdV-C5).

There have been over 100 human adenoviral serotypes described to date with differing entry receptors as well as prevalence rates within the human population [8]. HAdV-C5 has a particularly high prevalence in the human population, especially in Africa and Asia. Neutralising antibodies against HAdV-C5, generated in response to a natural pathogenic infection, may hamper the efficacy of HAdV-C5-based therapies when deployed clinically due to immune inactivation [9,10]. Alternative serotypes including HAdV-D10 have been explored to overcome the effects of anti-HAdV-C5 neutralisation [11]. In addition, HAdV-C5 engages coxsackie and adenovirus receptor (CAR) [12] as a primary means of cell entry, a receptor that is expressed both on erythrocytes and the tight junctions of epithelial cells (reviewed in [13]), yet is commonly downregulated by some types of cancer [14,15]. Furthermore, interactions between the major HAdV-C5 capsid protein, hexon and blood clotting factors, particularly factor (F) X, result in the rapid and efficient cellular uptake of HAdV-C5 virions via widely expressed heparan sulphate proteoglycans (HSPGs) [16,17,18]. This results in the depletion of the therapeutic effects due to widespread off-target infection and sequestration with inefficient tumour cell infection by HAdV-C5 [19]. HAdV-C5-based therapies for recurrent GBM are often delivered via intratumoral administration which limits off-target effects; however, these therapies have so far only been able to demonstrate modest success. Whilst demonstrating the safety and oncolysis via the intratumoral delivery of the virus, the overall median patient survival was not improved due to the standard of care (13 months) as the tumour invariably returned [20]. Local delivery is likely a necessity in GBM for overcoming the physical obstacle of the blood–brain barrier and to prevent local recurrence after surgical resection. We therefore investigated whether the development of Ad-based therapeutics using alternative adenovirus entry receptors may represent a more potent virotherapy for GBM. We have focused on HAdV-C5 vectors pseudotyped with fibre knob proteins derived from serotypes HAdV-D26, HAdV-B35 and HAdV-B3, which utilise the cell entry receptors sialic acid (SA) [21] or CD46 [22] and Desmoglein 2 (DSG2) [23], respectively. We demonstrate that GBM cell lines and tissue express all three adenoviral entry receptors tested, namely CAR, CD46 and DSG2. In addition, sialic acid is known to be expressed at high levels in the brain [24]. GBM does not express αvβ6 integrin, which is upregulated in many other types of aggressive cancers (e.g., ovarian, lung, skin and cervical cancer [25]), and is a promising target for novel adenoviral oncolytic therapies [11,26,27,28]. We also investigated the efficacy of various HAdV-C5 pseudotypes to gauge which adenoviral serotype may be better suited for oncolytic therapy in GBM. Our experiments revealed that both GBM and GBM stem cells express recognised adenoviral receptors and can be transduced by HAdV-C5 and HAdV-C5 with pseudotyped fibre knob proteins binding CD46, sialic acid and DSG2. Since these entry receptors are not unique to transformed GBM cells, and are expressed on normal cells, we additionally investigated whether improving selectivity, through the incorporation of tumour-specific promoters, could be employed to regulate the expression of potentially toxic or immunostimulatory transgenes in transformed GBM cells. Our findings indicate that a combination of alternative receptor usage, either through the use of pseudotyped or whole serotyped vectors, in combination with tumour-specific promoters such as survivin, may offer a powerful and highly selective means to deliver cytotoxic or immunostimulatory payloads selectively to GBM cells.

## 2. Materials and Methods

### 2.1. Cell Lines and Adenoviral Vectors

Human-patient-derived GBM stem cell lines L0 and L1 [29] were cultured as described previously [30,31]. Briefly, cells were cultured in DMEM-F12 supplemented with N2 (Thermo Fisher, Waltham, MA, USA, cat. no. 17502048) and 20 ng/mL EGF (R&D systems, Minneapolis, MN, USA cat. no. 236-EG-200), 20 ng/mL FGF (R&D systems Minneapolis, MN, USA, cat. No. 233-FB-010) and 100 μg/mL heparin (Sigma, Gillingham, UK cat. no. H4784). Cells were seeded at a density of 50,000 cells/mL, cultured as neurospheres and passaged every 7 days using Accutase (Sigma, Gillingham, UK cat. no. A6964) to disassociate the cells. E51 and E55 were obtained from the Glioma Cellular Genetics Resource (GCGR) from Professor Steve Pollard (University of Edinburgh). E51 and E55 were derived from GBM patients and cultured as previously described [32]. Cells were grown in DMEM-F12 (cat. no. D8437). Complete DMEM-F12 with 1.25% glucose (Sigma, Gillingham, UK G8644), 2% l MEM-NEAA (Gibco, Paisley, UK 11140-035), penicillin and streptomycin, 0.16% and 7.5% BSA solution (Gibco, Paisley, UK 15260-037), 0.2% beta mercaptoethanol (Gibco, Paisley, UK 31350-010), 1% B27 Supplement (Gibco, Paisley, UK 17504-044) and 0.5% N2 supplement (Gibco, Paisley, UK 17502-048) was used. Complete medium was supplemented with mouse EGF (10 ng/mL, peprotech, Cranbury, NJ, USA 315-09), human FGF (10 ng/mL, peprotech, Cranbury, NJ, USA 100-18b) and laminin (2 µg/mL, Sigma, Gillingham, UK, L2020). HFCAR cells were generated in house and cultured in DMEM supplemented with 10% FBS, 2% penicillin and streptomycin and 1% L-Glutamine.

Adenoviral vectors were produced as described previously using recombineering techniques [33]. Replication-deficient HAdV-C5 expressing a luciferase reporter gene provided the backbone for the replacement of fibre knob (K) protein to generate HAdV-C5/D26K, HAdV-C5/B35K and HAdV-C5/D49K [21]. HAdV-C5/B3K expressing the GFP reporter gene was provided by Professor Andre Lieber (University of Washington). HAdV-C5 vectors with luciferase expression under the control of the tumour-specific promoters hTERT and Survivin and Survivin/hTERT fusion promoter (Survivin-luciferase-BHG poly A-hTERT) were generated in house using recombineering techniques. Promoter sequences have been included in Appendix A. Viral vectors were expanded in 293-TRex cells and purified using two step CsCl gradient ultracentrifugation.

### 2.2. Flow Cytometry

Cells were incubated in FACS buffer (PBS) (Thermo Fisher, Waltham, MA, USA cat. no. 10010023) with 5% foetal bovine serum (Sigma, Gillingham, UK cat. no. F9665) and labelled with mouse primary antibodies: anti-αvβ6 10D5 (Millipore, Burlington, MA, USA cat no. mab20772), anti-CAR RmcB (Millipore, Burlington, MA, USA cat. no. 05-644), anti-CD46 258-MEM (Novus Biologicals, Littleton, CO, USA cat. no. LS-B5950-50) or anti-DSG2 CSTEM28 (Thermo Fisher, cat. no. 14-9159-82); all antibodies were diluted 1:500. Cells were subsequently stained with a secondary anti-mouse 647-conjugated antibody (Thermo Fisher, Waltham, MA, USA) and fixed with 4% paraformaldehyde. Appropriate fluorescence gating parameters were established using unstained and matched isotype control IgG-stained cells. In all of the experiments, doublets were eliminated using pulse geometry gates (FSC-H versus FSC-A). Single-cell suspensions were analysed using a BD AccuriTM C6 flow cytometer; FlowJo software (FlowJo LLC., Franklin Lakes, NJ, USA) was used for subsequent analyses.

### 2.3. Viral Infection and Luciferase Assays

Viral transduction assays were performed as described previously [34]. Briefly, cells were seeded at a density of 20,000 cells/well in a 96-microwell plate (Nunc, Thermo Fisher, Waltham, MA, USA) and infected with concentrations of 1000, 5000 or 10,000 virus particles (vps)/cell the following day. For each experimental condition, triplicate wells were treated, and a no-virus control was included. After 72 h, cells were lysed and the luciferase activity was quantified using a BioTek microplate reader and the luciferase assay system (Promega, Madison, WI, USA cat. no. E1500) according to the manufacturer’s instructions. In addition, total protein concentration per well was determined using a microplate absorbance plate reader (BioTek, Winooski, VT, USA) and the Pierce Micro BCA Protein assay (Thermo Fisher, Waltham, MA, USA cat. no. 10249133) to enable correction to relative luminescence units (RLUs) per mg of total protein. Viral infections with GFP-expressing vectors were carried out in the same manner; however, GFP levels were measured after 72 h via fixing cells in 4% paraformaldehyde and measuring fluorescence using the BD AccuriTM C6 flow cytometer, acquiring the data in the FL1 channel. Data were analysed using FlowJo software (FlowJo LLC.).

### 2.4. mRNA Expression Analysis

The GBM multiforme RNA-seq data from the Cancer Genome Atlas (TCGA) (*n* = 153) and normal brain Genotype Tissue expression (GTEx) (*n* = 1141) RNA-seq datasets were downloaded from the UCSC Xena RNA-seq Compendium (https://xena.ucsc.edu/), accessed on 3 March 2023, where the samples had been normalised and re-analysed using the same RNA-seq pipeline to eliminate batch effects [35]. The expression of CAR, CD46, DSG2, Survivin and hTERT was compared between glioblastoma tumour and normal brain samples.

### 2.5. Statistical Analysis

Statistical analyses were performed using Graphpad Prism 8 software (version 8.4.3). The normality of datasets was tested using the D’Agostino Pearson test and non-normally distributed data were compared using a Mann–Whitney U test. When comparing the means of ≥2 non-normally distributed groups involving one independent variable, the Kruskal–Wallis test with Dunn’s multiple comparisons was used. Statistically significant differences were marked as * *p* ≤ 0.05; ** *p* ≤ 0.01; *** *p* ≤ 0.001 and **** *p* ≤ 0.0001.

## 3. Results

### 3.1. Expression of the Adenoviral Receptors CAR, CD46 and DSG2 in GBM

The species C HAdV-C5 is well described as engaging CAR and a primary cell entry receptor, which anatomically is localised in tight junctions. Species B adenovirus serotypes HAdV-B35 and HAdV-B3 use the ubiquitously expressed membrane protein CD46 and the cell–cell adhesion protein DSG2, respectively. Using mRNA expression data taken from the TCGA dataset for GBM and the GTex dataset for normal healthy brain tissue, the presence of CAR, CD46 and DSG2 was compared (Figure 1A). CAR and CD46 mRNA were abundant in both healthy brains and GBM. CAR mRNA levels in GBM were significantly higher than in healthy brains. High levels of CD46 mRNA were observed in both healthy brains and GBM, although there were significantly higher mRNA levels in normal brains. DSG2 mRNA levels were substantially lower overall compared to CAR and CD46; however, there was greater expression of DSG2 mRNA in GBM compared to normal brain tissue. To conclude, although there is differential mRNA expression of adenoviral receptors between healthy brain tissue and GBM, the presence of native adenoviral receptors on normal brain tissue represents an obstacle when considering the development of oncolytic adenoviruses using these receptors as potential GBM treatments.

The expression of the adenoviral receptor on three GBM-derived cell lines was analysed using flow cytometry (Figure 1B). L1 was a suspension stem cell line derived from a high-grade GBM patient. E51 and E55 were adherent glioma stem cell lines derived from high-grade GBM patients. Figure 1B demonstrates receptor staining for CAR, CD46 and DSG2. L1 and E51 cells express high levels of all three adenoviral receptors. E55 cells were positive for CD46 but demonstrated low levels of both CAR (42.6%) and DSG2 (0.77%).

### 3.2. Transduction of Pseudotyped Adenoviral Vectors in GBM

GBM-derived cell lines L1, E51 and E55 were transduced using an HAdV-C5 vector and HAdV-C5 pseudotypes HAdV-C5/D26K, HAdV-C5/B35K and HAdV-C5/D49K, expressing the transgene under the control of a CMV IE promoter. L1 cells were transduced efficiently by all four viral vectors (Figure 2A). Interestingly, a dependent response was not observed in this cell line, suggesting that the threshold of infection was at the lowest dose of 1000 vps/cell. E51 cells were transduced by all viral pseudotypes, where we observed a dose-dependent response with increasing transduction from 1000 vps/cell to 10,000 vps/cell (Figure 2B). Both L1 and E51 express high levels of CAR and CD46; therefore, these transduction data are in line with the expected results. E55 cells express lower levels of CAR than L1 and E51; however, it remained permissive to transduction by HAdV-C5. All pseudotype vectors were able to transduce E55 cells (Figure 2C). Overall, after a comparison of all four viral vectors tested, HAdV-C5/D49K, which uses an unknown cellular receptor [36,37], transduced all three cell lines the most efficiently. This was also observed at a lower dose of 500 vps/cell used in L1 cells (Figure 2D). L1 cells were transduced with 500 vps/cell to determine whether this was under the threshold to observe a dose-dependent response. Interestingly, at this dose, it was apparent that HAdV-C5, HAdV-C5/D26K and HAdV-C5/B35K were not transduced as efficiently as HAdV-C5/D49K.

HAdV-B3 is recognised as engaging DSG2 for cell entry and therefore an HAdV-C5/B3K pseudotype vector expressing GFP as a reporter gene was used to investigate the differential usage of DSG2 via adenoviral vectors in GBM. The transduction of HAdV-C5/B3K was quantified via the expression of the GFP reporter gene in L1 (DSG2 high) and E55 (DSG2 low) cells. L1 cells demonstrated a dose-dependent transduction of the HAdV-C5/B3K virus; however, HAdV-C5 transduction remained consistent (approximately 60%) for all three doses (Figure 3A). HAdV-C5/B3K did not transduce L1 as efficiently as HAdV-C5, despite the presence of DSG2 on this cell line (Figure 1B). E55 cells were negative for DSG2 (Figure 1B), and consequently HAdV-C5/B3K did not transduce E55 cells efficiently even at the highest dose of 10,000 vps/cell (Figure 3B). Representative microscopic images were taken (Figure 3C). These data suggest that although there is a degree of receptor specificity, it cannot be relied upon for tumour-specific uptake into GBM cells. Post entry selectivity must also be considered to improve the safety of adenoviral vectors for the treatment of GBM.

### 3.3. Tumour-Specific Promoter hTERT and Survivin Offer Benefits in GBM Cells over Normal Cells

It has been reported that 75% of GBMs contain mutations in the human telomerase reverse transcriptase gene (hTERT) [38]. Similarly, survivin, an inhibitor of apoptosis, is expressed in nearly 80% of GBMs [39]. Adenoviral vectors that contain tumour-specific promoters exploiting the presence of hTERT and survivin may offer potential for the treatment of GBM. The TCGA dataset was used to evaluate the expression levels of these mRNA expressions for survivin and hTERT in GBM compared to normal brains (GTex). Figure 4A demonstrates that the mRNA expression of survivin is significantly higher in GBM compared to normal brain tissue. hTERT expression levels are also significantly higher in tumoral compared to healthy tissue (Figure 4B), supporting the reported literature. We therefore evaluated whether survivin or hTERT promoters demonstrated elevated selectivity of expression of a reporter gene in GBM-derived cells compared to human-derived fibroblasts expressing CAR (HF-CAR cells). Two GBM patient-derived cell lines L0 and L1, as well as HFCARs, were transduced using HAdV-C5 vectors containing luciferase under the control of either survivin, hTERT promoter or a fusion promoter that required both survivin and hTERT [40]. Cells were infected with 1000 vps/cell and luciferase activity was evaluated 72 h post infection (Figure 4C). HFCARs demonstrated levels of transduction that were barely above the background for all three adenoviral vectors. Conversely, when L0 and L1 were transduced efficiently by all three viral constructs containing tumour-specific promoters, HAdV-C5.survivin was the most efficient for both GBM cell lines. Interestingly, HAdV-C5.hTERT and HAdV-C5.hTERT/survivin demonstrated similar levels of luciferase activity, indicating that the combination of both hTERT and survivin did not improve the efficacy and may have hampered the effects when compared to survivin alone.

## 4. Discussion

GBM is a common and aggressive brain cancer with poor prognosis and significant unmet clinical needs. Oncolytic viruses have previously been evaluated for the treatment of GBM and a recombinant herpesvirus, Teserpaturev, has been licensed for the treatment of GBM in Japan [41]. Adenoviral vectors have demonstrated safety in clinical trials; however, the presence of native adenoviral receptors has yet to be described in GBM [20]. We have evaluated the expression of three well-described adenoviral receptors, CAR, CD46 and DSG2, in GBM compared to healthy brain tissue. The analysis of mRNA expression indicated that all three receptors were expressed in GBM and healthy brains; however, CAR and DSG2 were significantly upregulated in GBM. Adenoviral vectors such as HAdV-C5 and HAdV-B3 which use CAR and DSG2, respectively, could be utilised to target GBM via local delivery into the tumour. It is important to consider whether the lack of specificity may impact the healthy brain tissue surrounding the tumour.

We confirmed the expression of CAR, CD46 and DSG2 on the surface of patient-derived GBM stem cell lines L1, E51 and E55. Both L1 and E51 were positive for CAR and DSG2, whereas E55 expressed lower levels of CAR and was negative for DSG2. All three cell lines express high levels of CD46. These data are consistent with the mRNA expression levels reported in GBM. Native adenoviral receptors must be considered when engineering oncolytic adenoviral therapies for GBM.

We assessed the transduction of GBM using HAdV-C5-based pseudotype vectors. HAdV-C5 binds CAR via the engagement of the fibre knob protein. The fibre knob protein of HAdV-C5 was replaced with the fibre knob proteins of HAdV-D26 (known to use sialic acid), HAdV-B35 (binds CD46) and HAdV-D49 (receptor usage unknown [36,37,42]) to generate HAdV-C5-pseudotyped vectors. These vectors were then used to determine the transduction of these serotypes based on the fibre knob engagement. L1 cells were transduced consistently by all vectors at every concentration of the virus. When the concentration of the virus was reduced to 500 vps/cell, it was possible to determine that HAdV-C5/D49K demonstrated the highest transduction. This was also observed in E51 and E55 cells. Interestingly, HAdV-C5-pseudotyped vectors demonstrated improved transduction compared to HAdV-C5 at all concentrations of the virus. This may result from the availability of each receptor on the surface of the cells or additional factors such as engagement of the co-receptors αVβ3 and αvβ5 integrin that mediate internalisation. E55 cells express low levels of CAR; however, they demonstrate the similar transduction of HAdV-C5 when compared with E51, which suggests that there may be enough CAR present on the surface of these cells to mediate transduction. Differences between HAdV-C5 transduction in E51 and E55 may be discerned if a lower viral concentration is used. L1 and E55 cells were transduced with HAdV-C5 and HAdV-C5/B3K which engage DSG2 as a cell entry receptor [23]. HAdV-C5/B3K has previously been reported as demonstrating superior infectivity in glioma cells [43]. As previously demonstrated, HAdV-C5 was able to transduce both cell lines efficiently. HAdV-C5/B3K demonstrated a dose-dependent transduction in L1 cells; however, the transduction efficiency was lower than HAdV-C5. Additional work is required to determine whether this is due to viral receptor usage or the processing of the virus post entry. E55 cells were not efficiently transduced by HAdV-C5/B3K, consistent with the lack of DSG2 on this cell line. Considering the transduction data, HAdV-C5/D26K, HAdV-C5/B35K and HAdV-C5/D49K in particular appear to be superior candidates for the pseudotype oncolytic virotherapy vector for GBM compared to HAdV-C5 or HAdV-C5/B3K.

Finally, given the lack of selectivity of CAR, CD46 and DSG2 expression in GBM, we considered the need for additional methods of intrinsic cellular selectivity when designing oncolytic virotherapies targeting GBM. The use of a tumour-specific promoter, Glial fibrillary acidic protein (GFAP), has previously been described in a HAdV-C5 background and demonstrated selectivity towards glial tumours [44]. We evaluated the post entry selectivity of transgene expression mediated by tumour-specific promoters utilising the tumour markers hTERT and survivin. mRNA analysis confirmed that both hTERT and survivin were significantly upregulated in GBM compared to normal brain tissue; however, the difference was more marked in the case of survivin. We designed three adenoviral vectors, based on HAdV-C5, expressing luciferase under an hTERT promoter, survivin promoter and a fusion hTERT and survivin promoter [40]. These vectors were used to transduce GBM L0 and L1 cells as well as non-transformed human fibroblasts, HF-CARs. As expected, the adenoviral vectors were not able express luciferase in the HF-CAR cells due to the absence of hTERT and survivin. Both L0 and L1 cells were efficiently transduced by all three vectors; however, HAdV-C5.survivin was able to transduce these cells more efficiently than the HAdV-C5.hTERT and HAdV-C5.hTERT/survivin fusion, suggesting that survivin may be the most promising tumour-specific promoter for future therapeutic applications in GBM both in terms of the quantity and the selectivity of transgene expression in GBM cell lines. The levels of survivin and hTERT were not evaluated in these cells, and the levels could vary between patients; however, we have demonstrated that these vectors can selectively express transgenes in GBM cells. Future work will include developing therapeutics under the control of the survivin promoter for the treatment of GBM, potentially coupled as part of an HAdV-C5/D49K-pseudotyped vector to maximise the uptake and activity in GBM cells.

## 5. Conclusions

To conclude, we have evaluated the expression of well-described adenoviral vectors in GBM. We identified that CAR, CD46 and DSG2 expression in GBM and patient-derived GBM cells and GBM stem cells can be transduced by HAdV-C5-based adenoviral serotypes, in particular HAdV-C5/D49K. Given the lack of natural GBM selectivity of adenoviral receptors, we highlighted the need for additional selectivity and demonstrated that tumour-specific promoters can enhance the specificity of GBM cells, where the survivin promoter appears to be particularly well suited. These data inform the design of future oncolytic adenoviral therapies expressing therapeutic transgenes for the treatment of GBM.

## Figures and Tables

**Figure 1 viruses-15-01086-f001:**
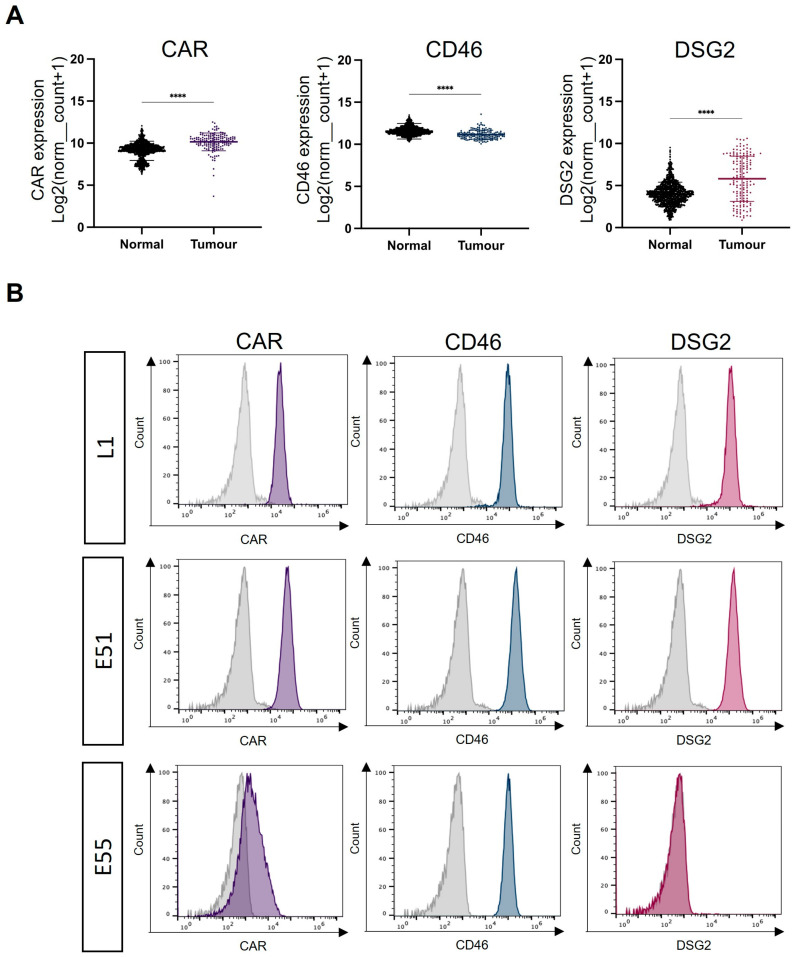
Expression of recognised adenoviral receptors on GBM, L1 cells and glioma stem cells (E51 and E55). (**A**) mRNA expression analysis of adenoviral receptors CAR, CD46 and DSG2 in GBM vs. normal brain. The data were collated from TCGA (tumour) and GTEx (normal) and represent *n* = 153 and *n* = 1141, respectively. Statistical significance was determined using the Mann–Whitney U test: **** *p* ≤ 0.0001. (**B**) Flow cytometry was used to determine receptor staining for CAR, CD46 and DSG2 on the surface of L1 (GBM) and glioma stem cells (E51 and E55). Histograms are representative examples from an individual experiment repeated in triplicate.

**Figure 2 viruses-15-01086-f002:**
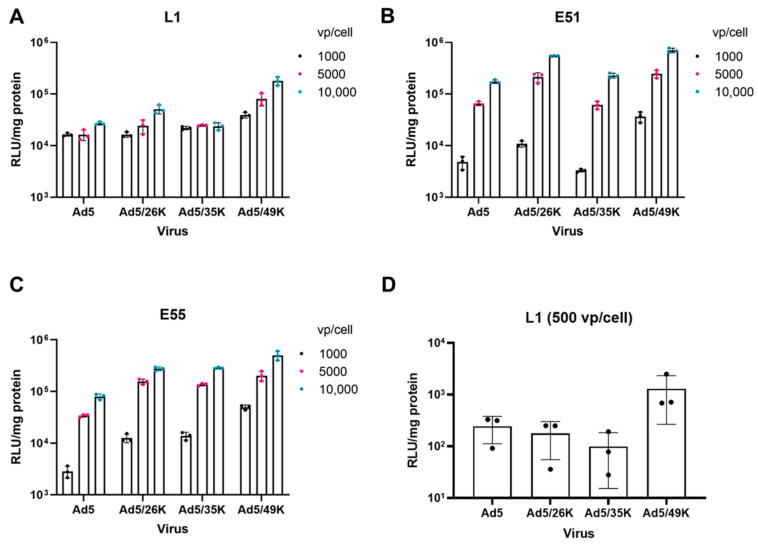
Transduction of L1, E51 and E55 cells via adenoviral pseudotype vectors. (**A**) L1 cells, (**B**) E51 cells and (**C**) E55 cells were transduced with 1000, 5000 and 10,000 vps/cell of adenoviral pseudotype vectors HAdV-C5 (Ad5), HAdV-C5/D26K (Ad5/26K), HAdV-C5/B35K (Ad5/35K) and HAdV-C5/D49K (Ad5/49K). (**D**) L1 cells were transduced with 500 vps/cell to investigate the threshold for infection. Luciferase activity was measured 72 h post infection. Data represent the mean of triplicates with individual data points shown. Error bars represent standard deviation, and data have been presented on a log scale.

**Figure 3 viruses-15-01086-f003:**
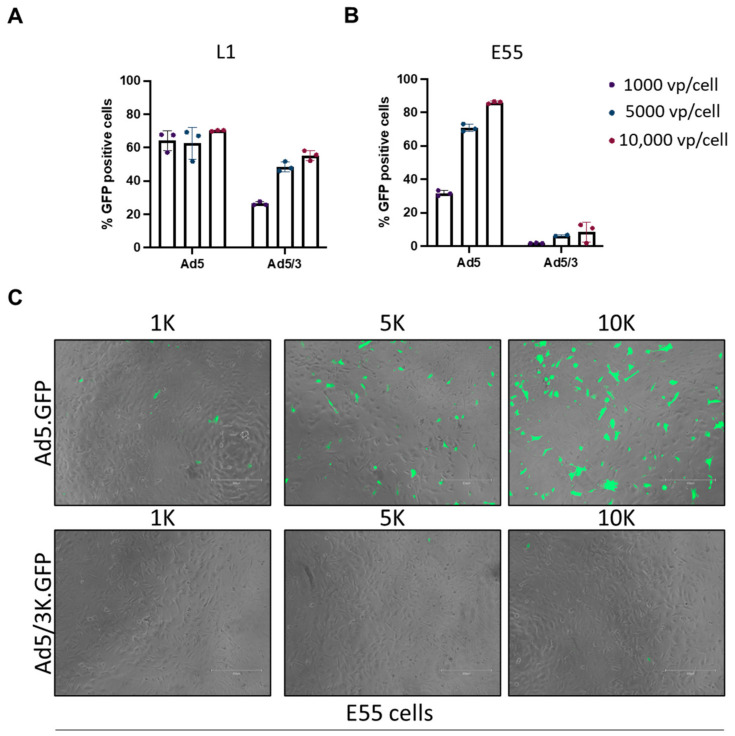
L1 and E55 cells were transduced with HAdV-C5 (Ad5) and HAdV-C5/B3K (Ad5/3) vectors expressing GFP. GFP levels were measured via flow cytometry and data were analysed using FlowJo. (**A**) L1 cells and (**B**) E55 were transduced with HAdV-C5.GFP and HAdV-C5/B3K.GFP at concentrations of 1000, 5000 and 10,000 vps/cell. Data are presented as mean of triplicate values of an independent experiment and individual data points are shown. (**C**) Representative images of E55 cells taken at 10X magnification bars using the EVOS cell imaging system (ThermoFisher). Images were taken both in TRANS and GFP channels and merged. Scale bar represents 300 µm.

**Figure 4 viruses-15-01086-f004:**
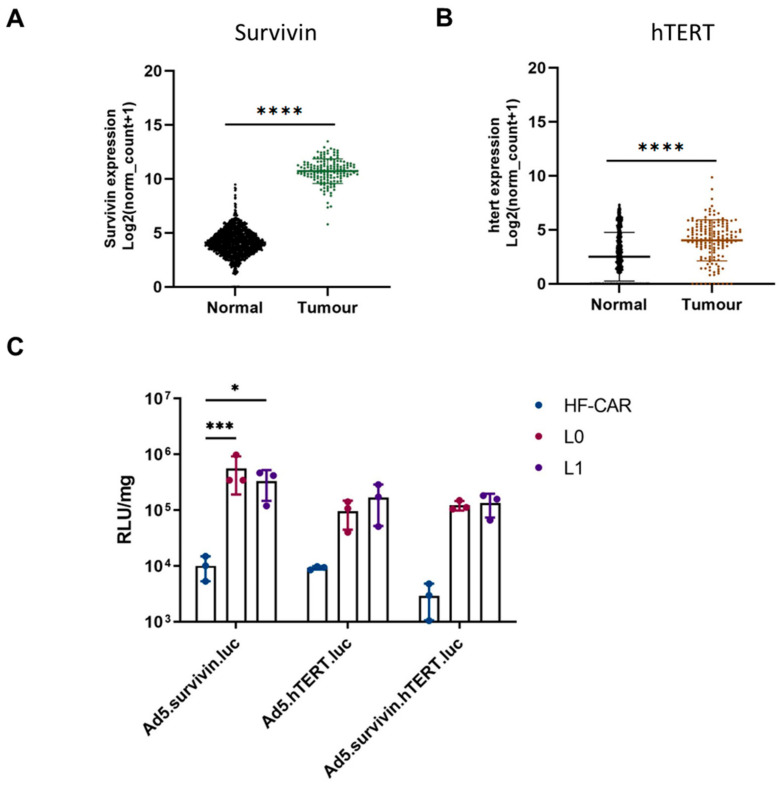
Evaluation of tumour-specific promoters hTERT and survivin. mRNA expression of (**A**) survivin and (**B**) hTERT were compared for GBM vs. normal brain. The data were collated from TCGA (tumour) and GTEx (normal) tissue and represent *n* = 153 and *n* = 1141, respectively. Statistical significance was determined using the Mann–Whitney U test: **** *p* ≤ 0.0001. (**C**) Adenoviral vectors based on HAdV-C5 (Ad5)-expressing luciferase under control of three different tumour-specific promoters: survivin, hTERT and a combination of survivin and hTERT. The virus was added to human-derived fibroblasts (HFCARs) and GBM-derived L0 and L1 cells at a concentration of 1000 vps/cell and luciferase activity was measured 72 h after infection. The data were plotted on a log scale and represent the mean of triplicate values. Statistical significance was determined using the Kruskal–Wallis test with Dunn’s multiple comparisons. Only significant values are shown: * *p* ≤ 0.05 and *** *p* ≤ 0.001.

## Data Availability

The data presented in this study are available upon request from the corresponding author. Publicly available datasets were also analysed in this study. These data can be found here https://xena.ucsc.edu/ accessed on 3 March 2023.

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
