# Peer review of "Engineering Adenoviral Vectors with Improved GBM Selectivity"

_viruses, 2023, doi:10.3390/v15051086_

Round 1

Reviewer 1 Report

The review of manuscript

Title: Engineering adenoviral vectors with improved GBM selectivity

·       Summary

In this manuscript, the authors reported a new approach for selective infection of adenovirus to glioblastoma (GBM). It is a valuable and critical method to increase to target GBM using engineered adenoviral vectors. In this manuscript, the authors well-organized data to report the cases and well-written the manuscript. However, because the data based on this paper can be a base line developing anti-cancer and the anti-cancer drug should be safe and have no or less side-effect to normal cells, the authors should carefully consider reviewing next questions.

·       Major issues

1.      There are no evidences about the engineered adenoviral vectors specifically infect to GBM only. The normal cells expressing DSG2 can become off-target for pseudotype vector. Are there any cells expressing DSG2 or becoming off-target? To solve this question, I recommend to add normal brain cells and HF-CAR cells in all figures.

2.      According to data, the expected target virus type is Ad5/3. But Ad3 cannot infect E55. So how the authors overcome this issue?

3.      The authors suggest the survivin has high potency to be a tumour specific promoter. In the figure 4, HF-CAR cells also express luciferase under the survivin promoter. Is the survivin promoter a good candidate for the tumour specific promoter?

4.      The resolution of figures is low. Make high resolution in X-axis and Y-axis. Do not just copy the figures.

Author Response

Summary

In this manuscript, the authors reported a new approach for selective infection of adenovirus to glioblastoma (GBM). It is a valuable and critical method to increase to target GBM using engineered adenoviral vectors. In this manuscript, the authors well-organized data to report the cases and well-written the manuscript.

Author response: We thank the reviewer for their commendation of our manuscript, recognizing the importance of the study and describing it as “well written” and the data as “well organized”.

However, because the data based on this paper can be a base line developing anti-cancer and the anti-cancer drug should be safe and have no or less side-effect to normal cells, the authors should carefully consider reviewing next questions.

Author response: We thank the reviewer for their helpful comments and address them below

  • Major issues
  1. There are no evidences about the engineered adenoviral vectors specifically infect to GBM only. The normal cells expressing DSG2 can become off-target for pseudotype vector. Are there any cells expressing DSG2 or becoming off-target? To solve this question, I recommend to add normal brain cells and HF-CAR cells in all figures.

Author response: We agree with the reviewer and we do not suggest that the vectors specifically target GBM only in this paper - indeed, Figure 1A demonstrates low levels of DSG2 in normal tissue. Instead, we propose that whilst pseudotyping may increase uptake of viral vectors in GBM, but that there would be additional post entry selectivity mechanisms needed such as tumour specific promoters. We have amended figure 4C to highlight the selectivity of expression in GBM using tumour specific promoters to cut the y axis off at 10e3. This demonstrates close to two logs improvement in selectivity of transgene expression in GBM cell lines compared to fibroblasts.

  1. According to data, the expected target virus type is Ad5/3. But Ad3 cannot infect E55. So how the authors overcome this issue?

Author response: We do not suggest Ad5/3 as a recommended therapeutic from the data in this paper. E55 do not express high levels of DSG2 and therefore are not able to be transduced by Ad5/3 which requires DSG2 for cell entry. Our paper simply summarises our observations on adenoviral receptor expression (CAR, CD46, DSG2) in GBM and normal tissues, provides data on the ability of viruses engaging these receptors in transducing GBM cells, and highlights that tumour specific promoters will be required to maximise on target activity if using adenoviral vectors based on their natural tropisms.

  1. The authors suggest the survivin has high potency to be a tumour specific promoter. In the figure 4, HF-CAR cells also express luciferase under the survivin promoter. Is the survivin promoter a good candidate for the tumour specific promoter?

Author response: We have amended the graph axis to better highlight the difference between Ad5 and Ad5-survivin. This is close to two logs, and we consider this a good candidate.

  1. The resolution of figures is low. Make high resolution in X-axis and Y-axis. Do not just copy the figures.

Author response: We have replaced the figures with higher resolution versions. Thanks for highlighting.

Reviewer 2 Report

The manuscript by Bates and coworkers describes the characterization of fiber-pseudotyped adenoviruses in glioblastoma cells. In addition, the authors assessed the potential of the hTERT and surviving promotor to drive transgene expression in GMB cells.

The work is well structured and clearly describe the experiments that were performed.

The manuscript could be improved is several ways.

11) The abstract state that oncolytic adenoviruses are a promising alternative treatment for GBM. However, the manuscript only describes the use of replication defective adenoviruses but not oncolytic adenoviruses. Please be consistent.

22)     There is little reference to prior work in the field. Others (e.g. DOI: 10.1038/sj.cgt.7701010) previously described similar experiments. Also the use of potential tumor-specific promoters in GBM has been demonstrated (DOI: 10.1002/jgm.1110). Please cover the exiting literature in more depth.

33)      Please follow the nomenclature of the adenoviruses a bit more strict (e.g. please amend Ad10 to read HAdV-D10, etc.). This would be helpful to the readers.

44)   The statement (@line 76) that attributes the modest success of HAdV-C5-based therapies in GBM to widespread off-target infection seems far-fetched since almost all studies employ intra-tumoral administration. Please correct.

Author Response

The manuscript by Bates and coworkers describes the characterization of fiber-pseudotyped adenoviruses in glioblastoma cells. In addition, the authors assessed the potential of the hTERT and surviving promotor to drive transgene expression in GMB cells.

The work is well structured and clearly describe the experiments that were performed.

Author response: We thank the reviewer for their positive review of the article and for describing is as “well structured” and “Clearly describing the experiments performed”.

The manuscript could be improved is several ways.

  • The abstract state that oncolytic adenoviruses are a promising alternative treatment for GBM. However, the manuscript only describes the use of replication defective adenoviruses but not oncolytic adenoviruses. Please be consistent.

Author response: The abstract refers to generally to oncolytic viruses as a therapeutic. Replication deficient viruses were used in the context of this study with a view of using oncolytic versions in the future.

  • There is little reference to prior work in the field. Others (e.g. DOI: 10.1038/sj.cgt.7701010) previously described similar experiments. Also the use of potential tumor-specific promoters in GBM has been demonstrated (DOI: 10.1002/jgm.1110). Please cover the exiting literature in more depth.

Author response: We thank the reviewer for highlighting these references. We have now cited both references: Targeting malignant gliomas with a glial fibrillary acidic protein (GFAP)-selective oncolytic adenovirus [45] on Line 354 and Human adenovirus type 35 vector for gene therapy of brain cancer: improved transduction and bypass of pre-existing an-ti-vector immunity in cancer patients [43] on Line 341.

3)      Please follow the nomenclature of the adenoviruses a bit more strict (e.g. please amend Ad10 to read HAdV-D10, etc.). This would be helpful to the readers.

Author response: We have changed the nomenclature of all adenoviruses throughout the manuyscript to the appropriate form, as suggested. The only exception is in the figures where the full nomelclature is too long for the axis, and we now mention the abbreviated for in the figure legend

4)   The statement (@line 76) that attributes the modest success of HAdV-C5-based therapies in GBM to widespread off-target infection seems far-fetched since almost all studies employ intra-tumoral administration. Please correct.

Author response: We corrected this statement on line 76

Round 2

Reviewer 1 Report

1. The revised form shows 'Ad5HADV-C5'. Is it an error or non-collected form? Please check again.

2. On the figure 4, there are old and new ones together. Please check again.